# The Influence of Flower Head Order and Gibberellic Acid Treatment on the Hydroxycinnamic Acid and Luteolin Derivatives Content in Globe Artichoke Cultivars

**DOI:** 10.3390/foods10081813

**Published:** 2021-08-05

**Authors:** María José Giménez, Marina Giménez-Berenguer, María Emma García-Pastor, Joaquín Parra, Pedro Javier Zapata, Salvador Castillo

**Affiliations:** 1Department of Food Technology, EPSO, University Miguel Hernández, Ctra. Beniel km. 3.2, 03312 Orihuela, Spain; maria.gimenezt@umh.es (M.J.G.); marina.gimenez02@goumh.umh.es (M.G.-B.); m.garciap@umh.es (M.E.G.-P.); pedrojzapata@umh.es (P.J.Z.); 2Agricultural Experimental Station of Elche (EEA/STT), CV-855, Ctra. Dolores km. 1, 03290 Elche, Spain; parra_joa@gva.es

**Keywords:** antioxidants, bioactive compounds, *Cynara cardunculus*, flower head weight, phenolic compounds

## Abstract

Flower head orders and the use of GA_3_ (gibberellic acid) treatment could be two influencing factors determining the bioactive compound levels in artichoke, but little to no information is available about their effects. In this study, we have therefore evaluated the influence of these factors on the hydroxycinnamic acid and luteolin derivative levels in three categories of artichoke: Seed-propagated open-pollinated cultivars; vegetatively propagated cultivars; and seed-propagated hybrids. The hydroxycinnamic acids and luteolin derivatives were quantified by RP-HPLC-DAD. The average flower head weight was the lowest in tertiary heads and GA_3_-treated artichokes, followed by secondary and main heads. Moreover, the hydroxycinnamic acid and luteolin derivatives levels were significantly higher in tertiary heads than in secondary or main heads. In addition, the GA_3_ treatment significantly reduced the hydroxycinnamic acid content and, in contrast, improved luteolin derivatives levels. These effects depended on the flower head order and cultivar. Knowledge of the effects of flower head order and GA_3_ treatment is therefore key in order to achieve the greatest health-benefits from artichoke consumption.

## 1. Introduction

Globe artichoke is a perennial plant belonging to the *Asteraceae* family. Artichoke is a rich source of antioxidant compounds and contains high amounts of total phenolic compounds [1], mainly caffeoylquinic acid derivatives and flavonoids such as luteolin glycosides. Caffeoylquinic acids are present in artichokes as mono-/di-caffeoyl esters, and the isomers of these acids have been described as the most abundant molecules [2,3]. Although the action mechanisms of the plant and its active principles are not fully known, caffeoylquinic acids and flavonoids appear to play major roles in its pharmacological properties [4]. The existence of phenolic compounds in the human diet correlates with a protective effect against certain chronic and degenerative diseases related to oxidative stress [5]. Polyphenols are able to reduce abnormally high levels of reactive oxygen species (ROS), which are found in a wide range of disorders, including inflammatory bowel disease and cancer [6]. Different preharvest factors have been found to influence the quantitative and qualitative profile of these secondary metabolites in the inflorescences, but several studies have shown that genetic material is a major determinant of artichoke quality [7,8].

The production of globe artichokes is largely based on vegetatively propagated clones of basal and lateral offshoots, as is the case in the ‘Blanca de Tudela’ cultivar. The most widely cultivated artichoke cultivar in Spain, ‘Blanca de Tudela’ has two production periods due to its reflowering ability: The first one coincides with the autumn-winter season (early production), and the second one, with the spring season (late production). ‘Blanca de Tudela’ has been characterised as an early cultivar. However, vegetatively propagated genotypes have several disadvantages, including susceptibility to pathogen transmission, physiologically heterogeneous plant material, a low rate of multiplication and limited flexibility in the transplant schedule, high planting costs and a high percentage of planting failures [9]. Different propagation methods have made it possible to improve crop yield, affecting earliness and head quality in order to meet consumer demand. Nevertheless, these disadvantages have prompted the development of new seed-propagated cultivars in the last three decades, beginning with open-pollinated cultivars, and later, hybrid cultivars [10]. Seed-propagated cultivars have several advantages, such as uniformity, high productivity, disease resistance, and profitability [11,12,13,14]. There is one drawback of using seed-propagated cultivars, however, which is their late entry into production due to a prolonged juvenile phase [15]. Growing from seedlings makes it possible to obtain high yields, however, even in regions with short vegetation periods [16].

In the same vein, the use of different chemical regulators has also been studied for controlling growth and earliness. The most widely used compound is gibberellic acid (GA_3_), which stimulates both cell elongation and division in addition to the development of lateral heads. The exogenous supply of GA_3_ is used to accelerate and synchronize flowering, promote early harvest, and enhance yield [10,17,18,19]. In new seed cultivars, the application of gibberellins can partially or totally replace the chill requirements to move the harvest forward to autumn. However, this treatment could have a negative effect on head size and weight and total yield since it boosts the production of secondary artichokes that are smaller than the primary ones [20]. The effectiveness of GA_3_ treatment is influenced by genotype, the sowing/planting date, concentrations added, and the frequency and number of applications [18,21,22,23]. Sharaf-Eldin et al. [24], for instance, observed that the timing of GA_3_ application was critical for decreasing time to flowering and altering the cynarin and chlorogenic acid levels in globe artichoke.

In our previous work [25], we studied the influence of flower head order on the phenolic content and quality of the ‘Blanca de Tudela’ cultivar. We found a significantly higher amount of total phenolics in tertiary heads, followed by secondary and main heads. No literature is available, however, on the influence of head order in other cultivars. The flower head order and GA_3_ treatment could be two influencing factors determining the bioactive compound content in artichoke. This important quality is related to the health benefits of artichokes and also influences their processing aptitude (for fresh consumption, canning or ready-to-eat processing, etc.). Therefore, the main aim of this research has been to study the influence of flower head order and GA_3_ treatment on the individual phenolic content of globe artichoke cultivars obtained by different propagation methods in order to fill this knowledge gap.

## 2. Materials and Methods

### 2.1. Plant Material and Experimental Design

This experiment was conducted during the 2018–2019 growing season on an outdoor experimental plot at the Elche Agricultural Experimental Station (Alicante; 38°14′53.06″ North, 0°41′48.38″ West). The experimental design was carried out in randomised block trials with two replicates for each cultivar and treatment. An experimental plot of 12 m^2^ for each cultivar was used, and nine artichoke plants were distributed at 1.67 × 0.8 m (Appendix A). The seeds were sown on 11 July, and the semi-dormant offshoots were planted on 27 July. Eight white artichoke genotypes with green heads and different biological and morphological profiles were selected and characterised (Figure 1): Two vegetatively propagated cultivars (‘Blanca de Tudela’ and ‘Calicó’) and six seed-propagated cultivars, including two open-pollinated cultivars (‘Lorca’ and ‘Tupac’) and four hybrids (‘Sambo’, ‘Symphony’, ‘Madrigal’, and ‘Olympus’). 

For the hormonal treatment, a commercial product was used in soluble concentrate form with a gibberellic acid concentration of 1.6% *w/v*. The doses applied were variable depending on the artichoke cultivar. ‘Lorca’ and ‘Olympus F1’ were treated with 30 ppm GA_3_ (1.875 cc L^−1^), and ‘Calicó’, ‘Shymphony F1’, ‘Madrigal F1’, and ‘Sambo F1’ were treated with 60 ppm (3.75 cc L^−1^). Gibberellic acid was applied every 15 days, when the plant had produced 7–8 true leaves, and a total of three treatments were applied to the plants during the crop cycle (17 September, 1 October, 15 October). No GA_3_ treatment was applied to the ‘Blanca de Tudela’ cultivar during the cycle due to its high degree of earliness. 

Crop management was performed according to the standard commercial practices used by growers in southeastern Spain. The harvest dates for artichokes treated with GA_3_ ranged from 14 November 2018 to 9 May 2019, with a total of 24 harvest dates along the entire crop cycle. For control or untreated artichokes, the harvest dates ranged from 4 December 2018 to 9 May 2019 with 22 harvest dates in total. Artichokes were harvested every 7–10 days following commercial export criteria, when flower heads reached their average commercial size. Next, the artichokes were classified according to the flower head order: Main, secondary, and tertiary heads. After harvest, the artichokes were transported to the laboratory and analysed on the same day. Ten artichokes per head order and cultivar were analysed individually.

### 2.2. Average Weight of Artichoke Flower Head Orders

Artichokes of each cultivar, flower head order, and treatment were harvested and weighed with 15–20 cm of the floral stem using a Radwag WLC 2/A2 Precision Balance (Radwag Wagi Elektroniczne, Poland) with accuracy to two decimal places. The average flower head weight was expressed as the mean ± SE in grams (g).

### 2.3. Extraction of Phenolic Compounds

For the phenolic extraction, 5 g of fresh receptacle was ground with 15 mL of methanol 80% containing 2 mM NaF for 2 min in an Ultra-Turrax^®^ (IKA TP 18) and then centrifuged at 10,000× *g* for 15 min at 4 °C. The homogenate was then filtered through 0.45 µm CHROMAFIL^®^ filters (Macherey-Nagel GmBh & Co., Düren, Germany), according to the method described by Martínez-Esplá et al. [26], for use in HPLC analysis.

### 2.4. Identification and Quantification of Hydroxycinnamic Acids and Luteolin Derivatives

The quantification of phenolic compounds was carried out using RP-HPLC-DAD. Phenolic extracts (20 µL) were analysed using a 1200 Infinity series HPLC coupled with diode array detection (Agilent Technologies, Waldbronn, Germany). The chromatographic separation was performed using a Luna C18 column (250 × 40 mm, 5 mm particle size, Tenokroma, Barcelona, Spain). The mobile phase was 1% formic acid in water (solvent A) and in acetonitrile (solvent B) with a flow rate of 1 mL min^−1^. The gradient started with 1% solvent to reach 15% B at 15 min, 30% at 30 min, 40% at 40 min, and 95% at 45 min, which was maintained to 50 min and then returned to 1% up until 60 min. Chromatograms were recorded at 320 and 360 nm for hydroxycinnamic acids and luteolins, respectively. Hydroxycinnamic acids were identified by their mass, spectra, and retention time, as can be seen in Appendix A, and the quantification was carried out according to previous findings reported by Martínez-Esplá et al. [26,27] using the 5-*O*-caffeoylquinic acid calibration curve. Luteolin derivatives were identified using the available literature [26,27] and internal standards purchased from Sigma-Aldrich (Norway, Germany). The quantification of luteolin derivatives were carried out using a calibration curve of quercetin 3-*O*-rutinoside, as it was reported previously [26,27]. An example of RP-HPLC-DAD chromatogram for luteolin derivatives is shown in Appendix A.

The following hydroxycinnamic acids were identified: 3-*O*-caffeoylquinic acid (3-CQA; t_r_ ≈ 7.3 min); 5-*O*-caffeoylquinic acid or chlorogenic acid (5-CQA; t_r_ ≈ 11.5 min); 1,3-di-*O*-caffeoylquinic acid or cynarin (1,3-diCQA; t_r_ ≈ 14.7 min); 3,4-di-*O*-caffeoylquinic acid (3,4-diCQA; t_r_ ≈ 22.2 min); 3,5-di-*O*-caffeoylquinic acid (3,5-diCQA; t_r_ ≈ 22.7 min); and 4,5-di-*O*-caffeoylquinic acid (4,5-diCQA; t_r_ ≈ 24.8 min). As for luteolin derivatives, the following three were identified: Luteolin 7-*O*-glucuronide 3-*O*-glucoside (Lut 7-gluc 3-glc; t_r_ ≈ 21.9 min); luteolin 7-*O*-glucuronide (Lut 7-gluc; t_r_ ≈ 23.5 min); and luteolin 7-*O*-glucoside (Lut 7-glc; t_r_ ≈ 23.8 min). The results for the major hydroxycinnamic acids and luteolins were expressed as g·kg^−1^, and the minor hydroxycinnamic acids were expressed as mg·kg^−1^ of fresh weight (FW). The total phenolic concentration was calculated as the sum of individual phenolics and was expressed as g·kg^−1^. Data are the mean ± SE.

### 2.5. Statistical Analysis

Data were subjected to the analysis of variance (ANOVA). The sources of variation detected were the artichoke head order and treatment. Mean comparisons were carried out using a multiple range test (Tukey’s HSD test) to find significant differences (*p* < 0.05) among artichoke head orders for each cultivar and treatment. For each cultivar and flower head order, significant differences at *p* < 0.05 between the treatments were analysed using Student’s *t*-test. All of the analyses were performed using the SPSS software package, version 22 (IBM Corp., Armonk, NY, USA).

## 3. Results and Discussion

### 3.1. Effect of Flower Head Order and Gibberellic Acid Treatment on Average Flower Head Weight

The seed-propagated open-pollinated cultivars, ‘Lorca’ and ‘Tupac’, showed significant differences in average weight among flower head orders not treated with GA_3_ (Table 1). The main heads of both cultivars had a 1.70-fold higher average weight than the secondary heads and showed an approximatley 2.20-fold increase with respect to the tertiary heads. In addition, the secondary heads of ‘Lorca’ and ‘Tupac’ had a 1.30- and 1.20-fold higher average weight, respectively, than the tertiary heads. GA_3_-treated artichokes maintained the significant differences among head orders for both cultivars, and the main heads showed the highest average weight, followed by the secondary and tertiary heads. The difference in average weight between secondary and tertiary heads was not observed in ‘Tupac’ artichokes, however, when they were treated with GA_3_. 

The vegetatively propagated cultivars, ‘Tudela’ and ‘Calicó’, also showed significant differences among flower head orders (Table 1). ‘Tudela’ artichokes were not treated with GA_3_, and they showed a 1.16-fold higher average weight in the main heads than in the secondary heads, and in the secondary heads than in the tertiary heads. Similar results were obtained in the ‘Calicó’ cultivar. The GA_3_ treatment influenced the average weight of the tertiary heads in this cultivar, however, in the GA_3_-treated ‘Calicó’ artichokes, the average weight of the tertiary heads was 1.46-fold lower than in the untreated samples.

‘Sambo’, ‘Symphony’, ‘Madrigal’, and ‘Olympus’ showed a similar behaviour, displaying significant differences among flower head orders both in treated and untreated artichokes (Table 1). These seed-propagated hybrids had also a significantly higher average weight in the main heads, followed by the secondary and tertiary heads. However, the influence of the GA_3_ treatment was cultivar-dependent. In the ‘Sambo’ and ‘Madrigal’ cultivars, for instance, the average weight of the main, secondary, and tertiary flower head orders was significantly influenced by the treatment. For these cultivars, the average weight of the untreated flower head orders was significantly higher than that of the artichokes treated with GA_3_. In ‘Symphony’ and ‘Olympus’ artichokes, the treatment only affected the average weight of the main and tertiary heads, respectively. Thus, the ‘Symphony’ main heads and the ‘Olympus’ tertiary heads showed a 1.61- and 1.32-fold higher average weight, respectively, in artichokes not treated with GA_3_.

Around 60% of world globe artichoke production is for the fresh market. Generally, the standard weight of heads ranges between 200–500 g (and can reach up to 700 g), but smaller ones can often be found (weighing 150–200 g) [28]. Heads are sold with a 4–6-cm long peduncle. This crop is of great interest to the agro-industry for processes such as canning and freezing [15,29]. Among the genotypes tested in this study, the vegetatively propagated cultivar, ‘Calicó’, and the seed-propagated hybrids, in particular ‘Sambo’ and ‘Madrigal’, stand out compared to the seed-propagated open-pollinated types. In fact, these cultivars had the largest heads in terms of both size (Figure 1) and, especially, weight (Table 1). Our average head weight results for some cultivars, such as ‘Madrigal’, agree with the weights reported by Bonasia et al. [30].

As far as we know, this is the first time that the influence of flower head orders on average weight has been reported in artichoke. In all of these cultivars studied, the main head orders showed a significantly higher average weight than the secondary and tertiary heads, and the tertiary heads were significantly smaller than the secondary ones (Table 1). In the case of ‘Tupac’ artichokes, however, no significant difference in average weight was observed between secondary and tertiary heads when this cultivar was treated with GA_3_. 

Gibberellic acid (GA_3_) greatly affects the vernalisation of artichokes when applied to plants with 10 or more true leaves [31]. In addition, GA_3_ influences the leaves initiation and stem elongation, as well as the development of lateral heads [21,32]. In the present work, the GA_3_ treatment negatively affected the average weight of ‘Calicó’, ‘Sambo’, ‘Madrigal’, ‘Symphony’, and ‘Olympus’ cultivars. Specifically, this treatment significantly reduced the average weight of the tertiary heads in the vegetatively propagated cultivars studied and the main, secondary or tertiary heads in the seed-propagated hybrids, depending on the cultivar. The GA_3_ treatment significantly decreased the average weight of tertiary heads in most of the seed-propagated hybrids studied, with the exception of the ‘Symphony’ cultivar. Although applying GA_3_ to artichokes primarily affects flower initiation and early harvest, it can have a negative impact on head size, weight, and total yield [21,31,33]. For the first time, we have shown that the application of GA_3_ has a great influence on the average weight of some flower head orders, but this influence directly depends on the cultivar (Table 1). In sum, the average head weight was significantly influenced by the flower head order and GA_3_ treatment, and both factors were cultivar-dependent.

### 3.2. Effect of Flower Head Order and Gibberellic Acid Treatment on Hydroxycinnamic Acid Content

The 5-CQA is the main hydroxycinnamic acid in the artichoke (Figure 2), followed by 3,5-diCQA (Figure 3) and 3,4-diCQA (Figure 4). All of the cultivars studied, with or without the GA_3_ treatment, showed significant differences in terms of 5-CQA levels among flower head orders (Figure 2A,B). The 5-CQA content was significantly higher in the tertiary heads than in the main and secondary heads in all cultivars analysed, with the exception of untreated ‘Lorca’ and ‘Symphony’ artichokes, in which no significant differences were observed. In ‘Tudela’, the main heads had 1.16- and 1.29-fold less 5-CQA than the secondary and tertiary heads, respectively, with the tertiary flower heads showing the highest 5-CQA content. Similar results were observed in ‘Olympus’ artichokes treated with GA_3_. The influence of this treatment on the 5-CQA content in flower head orders depended on the cultivar (Figure 2A). In ‘Lorca’ and ‘Tupac’, for instance, levels of this major hydroxycinnamic acid in the main heads of GA_3_-treated artichokes were 1.42- and 1.23-fold lower, respectively, than in untreated artichokes. No significant differences in the 5-CQA content were observed between both treatments for ‘Calicó’ and ‘Sambo’ cultivars, however. In the ‘Symphony’ and ‘Olympus’ cultivars, on the other hand, levels of this major phenolic compound in the main, secondary, and tertiary heads were significantly influenced by the GA_3_ treatment. In these cultivars, the GA_3_-treated flower head orders showed significantly lower 5-CQA levels than the untreated heads. Similar results were observed in ‘Madrigal’ artichokes, with the exception of the main heads, whose 5-CQA content was not affected by GA_3_.

The 3,5-diCQA content was significantly higher in the tertiary heads of most of the cultivars studied, whether treated with GA_3_ or untreated (Figure 3A,B). However, some cultivars, such as ‘Lorca’ (without treatment) and ‘Calicó’ (with and without treatment), did not show significant differences in 3,5-diCQA among heads. The tertiary heads of treated ‘Lorca’ artichokes and untreated ‘Tudela’, ‘Sambo’, and ‘Madrigal’ artichokes showed significantly higher 3,5-diCQA levels than the secondary heads, and the levels in the latter were significantly higher than in the main heads. On the other hand, the GA_3_ treatment significantly reduced the 3,5-diCQA content in the main, secondary, and tertiary heads of ‘Calicó’ and ‘Sambo’ cultivars compared to the untreated heads (Figure 3A). In ‘Madrigal’, secondary and tertiary heads in GA_3_-treated artichokes showed 1.55- and 1.29-fold less of this hydroxycinnamic acid, respectively, than the untreated heads, while the GA_3_ treatment only influenced the tertiary heads in ‘Lorca’, ‘Tupac’, and ‘Olympus’ artichokes. Tertiary heads in ‘Lorca’ showed a 1.36-fold increase in hydroxycinnamic acid levels and ‘Tupac’ and ‘Olympus’ a 1.10-fold decrease.

The 3,4-diCQA content showed significant differences among flower head orders for each cultivar, except for ‘Lorca’ (without treatment), ‘Madrigal’ and ‘Symphony’ (with treatment), and ‘Calicó’ (with and without treatment) (Figure 4A,B). Specifically, the 3,4-diCQA content in tertiary heads of the cultivars that showed significant differences was significantly higher than in the main heads. ‘Tudela’ was the only cultivar that showed a 1.19-fold increase in 3,4-diCQA content between the main and secondary heads and a 1.22-fold increase from the secondary to the tertiary heads. The GA_3_ treatment significantly reduced the 3,4-diCQA content in both the secondary and tertiary heads of the ‘Sambo’ and ‘Madrigal’ cultivars (Figure 4A). On the other hand, this treatment only affected the 3,4-diCQA content in tertiary heads of ‘Lorca’, ‘Tupac’, ‘Symphony’, and ‘Olympus’, with ‘Lorca’ showing a 2.94-fold increase and ‘Tupac’, ‘Symphony’, and ‘Olympus’ showing decreases of 1.77-, 1.62-, and 1.75-fold, respectively, compared to the untreated flower heads.

The amounts of the following minor hydroxycinnamic acids were also determined in the flower head orders: 3-CQA, 1,3-diCQA, and 4,5-diCQA. Data are shown in Figure 5 (Appendix A), which depicts the biosynthesis pathway among the major and minor hydroxycinnamic acids in artichoke. The 3-CQA is obtained from 5-CQA, as the main hydroxycinnamic in artichoke, by an unknown enzyme. As a general trend, the content levels of this individual phenolic compound were significantly higher in tertiary heads than in main heads, except in the GA_3_-treated ‘Calicó’ and ‘Sambo’ artichokes and untreated ‘Tupac’ artichokes. ‘Tudela’, GA_3_-treated ‘Tupac’, and untreated ‘Olympus’ showed significant differences between heads, and the 3-CQA content increased from the main to the tertiary heads. In the ‘Tupac’ and ‘Symphony’ cultivars, the GA_3_ treatment mainly increased the 3-CQA content in the three flower head orders. 

On the other hand, 1,3-diCQA is derived from 3,5-diCQA by an unknown enzyme. Most of the cultivars studied, ‘Lorca’, ‘Tupac’, ‘Sambo’, ‘Symphony’, ‘Madrigal’, and ‘Olympus’, showed significant differences in the 1,3-diCQA content between the three flower head orders. In these cultivars, the 1,3-diCQA content was significantly higher in untreated artichokes and in the tertiary heads than in the main and secondary heads. The 4,5-diCQA is derived from the same biosynthesis pathway as 1,3-diCQA from the biosynthesis of 3,5-diCQA mediated by the CQT enzyme (hydroxycinnamoyl-CoA quinate cinnamoyl transferase). No significant differences were observed in the 4,5-diCQA content among flower head orders of the ‘Calicó’, ‘Sambo’, and ‘Symphony’ cultivars treated with GA_3_. The 4,5-diCQA content was significantly higher in tertiary heads than in secondary heads, however, and the secondary heads showed more 4,5-diCQA than the main heads in the GA_3_-treated ‘Olympus’ artichokes and in the untreated ‘Tupac’ and ‘Tudela’ artichokes. The GA_3_ treatment significantly reduced the 4,5-diCQA levels in the three flower head orders of the ‘Tupac’, ‘Calicó’, and Madrigal’ cultivars.

Artichoke is highly abundant in polyphenols, especially hydroxycinnamic acid derivatives, which represent a significant fraction of the whole secondary metabolites. In agreement with several studies performed on artichoke crops [8,34,35,36,37], the 5-CQA or chlorogenic acid was the most abundant compound within the caffeoylquinic derivatives evaluated in the artichoke cultivars in the present study (Figure 2). It is important to note that, together with 5-CQA, 3,5-diCQA (Figure 3) and 3,4-diCQA (Figure 4) were the main phytochemicals quantified in all of the artichokes among the hydroxycinnamic acids, as previously reported by Martínez-Esplá et al. [26]. Accordingly, the minor hydroxycinnamic acids determined in these eight artichoke cultivars were 3-CQA or neochlorogenic acid, 1,3-diCQA or cynarin and 4,5-diCQA (Figure 5). According to our results, the hydroxycinnamic acid content varied among flower head orders in most of the artichoke cultivars studied, and the tertiary heads showed higher levels of the major and minor hydroxycinnamic acids than the main or secondary heads. The individual phenolic content of the flower head orders varied widely among the eight artichoke cultivars studied. For example, the untreated ‘Lorca’ cultivar did not show significant differences in any of the hydroxycinnamic acids analysed among the three flower head orders. ‘Tudela’ tertiary heads, on the other hand, showed significantly higher amounts of the three major hydroxycinnamic acids than the secondary and main heads, with the latter showing the lowest individual phenolic content.

This variability in caffeoylquinic acid derivative levels in artichoke heads has also been reported in relation to different harvest times represented by the different classes of heads (main, secondary, and processing) in the ‘Violetto di Provenza’ cultivar [38]. In this study, 5-CQA or chlorogenic acid in the processing head class was about 2-fold higher than in the main head class. The existence of several isozymes of quinate hydroxycinnamoyl transferase (HQT) in artichoke could potentially explain the possible fluctuations in chlorogenic acid levels in the selected cultivars [39]. Our findings also agree with our previous work performed on the commercial cultivar ‘Blanca de Tudela’ for evaluating variations in polyphenolic composition [25]. The results showed that there were differences in phenolic compound levels as well as in the size and appearance of the artichoke heads throughout the production period and depending on the irrigation regimes applied. The main, secondary, and tertiary head orders of this cultivar showed significant differences in hydroxycinnamic acid levels, and the levels were highest in the tertiary heads, followed by secondary and main head orders. The current study highlights the fact that tertiary head orders have more individual hydroxycinnamic acids than secondary or main artichoke heads, and that this influence depends on the cultivar analysed. 

With regards to the effect of treatment, GA_3_ decreased the levels of all hydroxycinnamic acids compared to the levels found in untreated artichokes, with the exception of two minor phenolic compounds—3-CQA and 1,3-diCQA—whose levels slightly increased in the treated head orders (Figure 5). These results are in accordance with those reported by Sharaf-Eldin et al. [24]. In this research work, when plants that were directly seeded into the field were analysed, the highest chlorogenic acid and cynarin levels were found in the control or untreated artichoke plants. The GA_3_ treatments also decreased the phenolic content in the edible portion of the heads or receptacles. When the average phenolic content of all of the flowers harvested were analysed, it is clear that more phenolics were produced in the transplanted artichokes that had not been treated with GA_3_. These authors concluded that although the GA_3_ treatment increased chlorogenic acid in artichoke leaves, it decreased chlorogenic acid and cynarin levels in the flower receptacles, regardless of the planting procedure used. Therefore, the variation in hydroxycinnamic acid content among cultivars could also be due to the timing of GA_3_ application and a complex metabolic process occurring during the shift from vegetative to reproductive growth.

### 3.3. Effect of Flower Head Order and Gibberellic Acid Treatment on the Luteolin Derivatives Content

Three luteolin derivatives were quantified in the artichoke cultivars studied, as can be seen in Figure 6: Lut 7-gluc, Lut 7-gluc 3-glc, and Lut 7-glc. Lut 7-gluc was the major luteolin identified in the cultivars studied, followed by Lut 7-glc and Lut 7-gluc 3-glc. The Lut 7-gluc content was significantly higher in the tertiary heads than in the secondary or main heads, and this influence was cultivar dependent. Specifically, ‘Tudela’, treated ‘Madrigal’ and untreated ‘Lorca’, ‘Tupac’, ‘Sambo’, ‘Symphony’, ‘Madrigal’, and ‘Olympus’ cultivars showed significantly higher Lut 7-gluc levels in the secondary heads than in the main heads and in the tertiary heads than in the secondary heads. In contrast to the hydroxycinnamic acid results, the Lut 7-gluc content was improved by the GA_3_ treatment. Lut 7-gluc 3-glc levels were also enhanced by the GA_3_ treatment, especially in the main, secondary, and tertiary heads of ‘Sambo’ and ‘Olympus’ cultivars. The untreated ‘Lorca’, ‘Tudela’, ‘Symphony’, and ‘Olympus’ showed significant differences in Lut 7-gluc 3 glc content among their flower head orders, with the tertiary heads showing the highest levels, followed by the secondary and main heads. Artichokes treated with GA_3_ showed significantly higher Lut 7-glc levels than the untreated artichokes, while treated ‘Calicó’ and ‘Olympus’ and untreated ‘Tupac’, ‘Tudela’, ‘Calicó’, ‘Sambo’, ‘Symphony’, and ‘Olympus’ all showed the highest Lut 7-glc levels in their tertiary heads, followed by the secondary and main heads. The main, secondary, and tertiary heads of ‘Calicó’ and ‘Olympus’ were significantly influenced by the treatment, with all of the treated flower head orders showing a significant increase in Lut 7-glc levels.

Lut 7-gluc was the major luteolin identified in the artichoke cultivars, followed by Lut 7-glc and Lut 7-gluc 3-glc, as has been reported by other authors [26,38]. The total amounts of each individual luteolin were significantly higher in tertiary flower head orders than secondary or main heads. However, genetic variability among the studied cultivars highly influenced the average content of each luteolin derivative in the flower head orders. In our previous preliminary work, similar individual luteolin derivative levels were obtained among flower head orders in the ‘Blanca de Tudela’ cultivar [25]. No literature is available related to the influence of flower head orders on luteolin derivative levels in other artichoke cultivars. Therefore, our results fill this knowledge gap for the first time in the white globe artichoke cultivars analysed. 

Therefore, the GA_3_ treatment had an opposite effect on luteolin derivatives levels compared to hydroxycinnamic acid levels. The average Lut 7-gluc, Lut 7-glc, and Lut 7-gluc 3-glc content was improved by the GA_3_ treatment. As far as we know, no literature is available concerning the effect of GA_3_ on luteolin derivatives content. Nevertheless, Gagliardi et al. [38] reported that the concentrations of luteolin (Lut 7-gluc and Lut 7-glc) were significantly influenced by wastewater irrigation. In particular, secondary and tertiary municipal wastewater significantly increased the major luteolin present in artichoke extracts (Lut 7-gluc). Accordingly, the treatment with oxalic acid (OA) at 2 mM led to higher luteolin concentrations both at harvest and during cold storage in ‘Blanca de Tudela’ artichokes [26]. These results are relevant since some flavones are not distributed widely in food plants, and globe artichoke represents a significant dietary source of these compounds and their conjugates [8].

### 3.4. Effect of Flower Head Order and Gibberellic Acid Treatment on Total Phenolic Content

The total phenolic content showed significant differences among flower head orders and was also cultivar-dependent (Appendix A). ‘Tudela’ was the only cultivar that showed a 1.30-fold increase in total phenolic content between secondary and main head orders and a 1.17-fold higher phenolic content in tertiary heads than in secondary ones. As a general trend, all of the cultivars had a higher total phenolic content in tertiary flower head orders than in secondary or main ones. In those studied cultivars, excluding ‘Tudela’, no significant differences were observed between main and secondary head orders or secondary and tertiary heads. The GA_3_ treatment only influenced the total phenolic content of ‘Sambo’ and ‘Madrigal’ seed-propagated hybrids. Specifically, the total phenolics in the secondary head orders of these two cultivars were significantly reduced by the GA_3_ treatment, by 1.30-fold. In addition, the total phenolic content also decreased by 1.39- and 1.18-fold in the tertiary head orders of the ‘Sambo’ and ‘Madrigal’ cultivars, respectively.

The total phenolic content, calculated as the sum of individual phenolic compounds, varied among genotypes, with ‘Tudela’, ‘Sambo’, ‘Symphony’, and ‘Madrigal’ displaying the highest levels (≈4.50–9.00 g·kg^−1^ FW), and ‘Tupac’ and ‘Olympus’ showing the lowest levels (≈2.50–6.50 g·kg^−1^ FW). Our results concerning the total phenolic content of the ‘Symphony’ cultivar agree with the results of Turkiewicz et al. [40], who found that this cultivar accumulated the greatest amounts of phenolic acids (3.44 g·kg^−1^ DW). In the present work, however, ‘Sambo’ artichoke showed higher total phenolic levels than those reported by Turkiewicz et al. [40]. These differences in values obtained could be due to many factors, such as climatic conditions, agrotechnical techniques or maturity stage [40]. Basically, the main phenolic acids detected in the globe artichoke cultivars studied were caffeoylquinic acid, particularly chlorogenic acid (5-CQA), and di-caffeoylquinic acids, as well as flavones such as luteolin derivatives (both glucuronides and glycosides). These results are consistent with other studies [8,34,35,40,41,42,43]. These polyphenolic compounds have a marked scavenging activity against ROS and free radicals, acting as a protective pool against oxidative damage to biological molecules such as protein and lipids [44].

As seen in the present study and observed previously [7,8,43], the total phenolic content varies significantly among cultivars and in relation to the different parts of the plants. The ‘Tudela’ cultivar was the only cultivar that showed a significant difference in total phenolic content among flower head orders. In this cultivar, the tertiary flower heads had significantly higher levels of phenolic compounds than the secondary and main heads, and these main heads showed significantly lower levels than the secondary ones. Accordingly, the same influence of flower head order on total phenolic content was previously reported by Giménez et al. [25]. For the first time, our results demonstrate that the tertiary flower heads of the white artichoke cultivars analysed, excluding ‘Tudela’, present a higher polyphenolic content than the secondary or main heads, and that this influence is cultivar-dependent. Moreover, the GA_3_ treatment greatly influenced the total phenolic compound content in the secondary and tertiary head orders of the two seed-propagated hybrids characterised in the present study, ‘Sambo’ and ‘Madrigal’. The decrease in this total content after using the GA_3_ treatment to promote an early harvest has important implications for marketing and is extremely interesting to note if we consider the artichoke as a functional food according to the European Commission on Functional Food Science in Europe (FuFoSe) [45]. 

Indeed, phenolic compounds are reported to have beneficial effects in the treatment of hepatobiliary diseases, hyperlipidaemia, dropsy, rheumatism, and cholesterol metabolism [35]. These broad therapeutic indications cannot be ascribed to a single compound, but to several active compounds providing additive or synergistic pharmacologic effects, including mono-caffeoylquinic and di-caffeoylquinic acids and flavonoids such as luteolin derivatives [34,46,47]. 

Similar results were obtained in a previous study in which the edible receptacles of flower heads produced by control artichoke plants had a higher phenolic content than those of the GA_3_-treated plants [24]. These authors concluded that the GA_3_ treatments decreased the phenolic content in the edible portion of the ‘Imperial Star’ heads. Our results highlight the fact that the functional quality of artichokes is strongly influenced by genotype, with high variability in individual phenolic content influenced by two main factors: Flower head orders and gibberellic acid treatment.

## 4. Conclusions

Our results report, for the first time, that the levels of individual phenolic compounds (specifically hydroxycinnamic acids and luteolin derivatives) in seed-propagated open-pollinated, vegetatively propagated, and seed-propagated hybrid globe artichoke cultivars are highly influenced by both the flower head order and gibberellic acid treatment. Firstly, tertiary head orders showed the highest individual phenolic content, followed by secondary and main heads, and this effect was cultivar-dependent. Secondly, the gibberellic acid treatment affected the content of these bioactive compounds in different ways. Artichokes treated with gibberellic acid showed lower hydroxycinnamic acid concentrations than the untreated artichokes, but greater luteolin derivative content in those gibberellic acid-treated artichokes levels. The influence of gibberellic acid depended on the flower head order and artichoke cultivar. 

In conclusion, the results of this study are an attempt to fully characterise the individual phenolic influence in certain globe artichoke cultivars. Our aim is to contribute knowledge to the study of the functional aspects of this important traditional plant food and to gain a better understanding of their potential health-promoting properties. Moreover, this study could be of use for improving the use of gibberellic acid treatments in artichoke crops so that the inflorescences have greater beneficial effects for human health.

## Figures and Tables

**Figure 1 foods-10-01813-f001:**
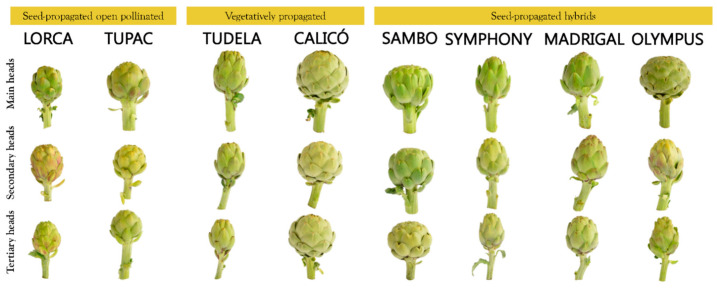
Main, secondary, and tertiary heads of seed-propagated open-pollinated cultivars (‘Lorca’ and ‘Tupac’), vegetatively propagated cultivars (‘Blanca de Tudela’ and ‘Calicó’), and seed-propagated hybrids (‘Sambo’, ‘Symphony’, ‘Madrigal’, and ‘Olympus’).

**Figure 2 foods-10-01813-f002:**
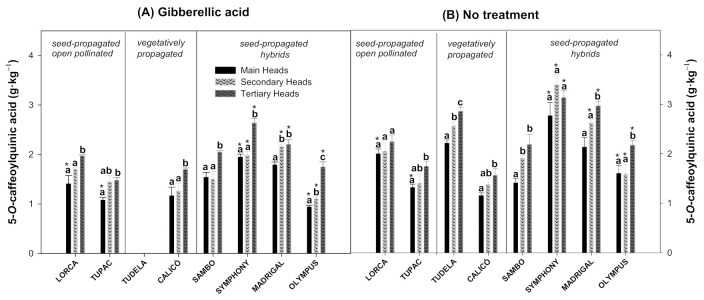
The 5-*O*-caffeoylquinnic acid content (g·kg^−1^) of globe artichoke cultivars influenced by flower head order (main, secondary, and tertiary heads) and treatment (gibberellic acid (**A**) or no treatment (**B**)). Data are the mean ± SE. Different letters show significant differences (*p* < 0.05 according to HSD Duncan’s test) among flower head orders for each artichoke cultivar and treatment. Significant differences between both treatments (*p* < 0.05 according to Student’s *t*-test) were expressed as * placed in each flower head order for each artichoke cultivar.

**Figure 3 foods-10-01813-f003:**
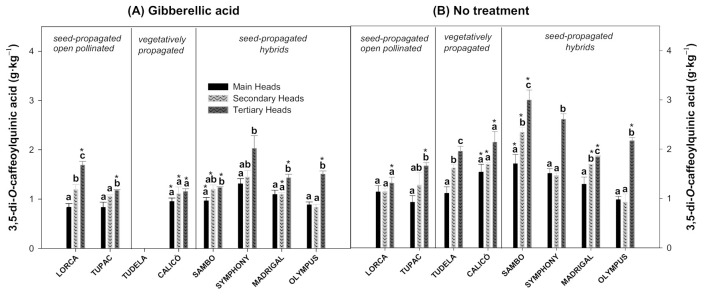
The 3,5-di-*O*-caffeoylquinnic acid content (g·kg^−1^) of globe artichoke cultivars influenced by flower head order (main, secondary, and tertiary heads) and treatment (gibberellic acid (**A**) or no treatment (**B**)). Data are the mean ± SE. Different letters show significant differences (*p* < 0.05 according to HSD Duncan’s test) among flower head orders for each artichoke cultivar and treatment. Significant differences between both treatments (*p* < 0.05 according to Student’s *t*-test) were expressed as * placed in each flower head order for each artichoke cultivar.

**Figure 4 foods-10-01813-f004:**
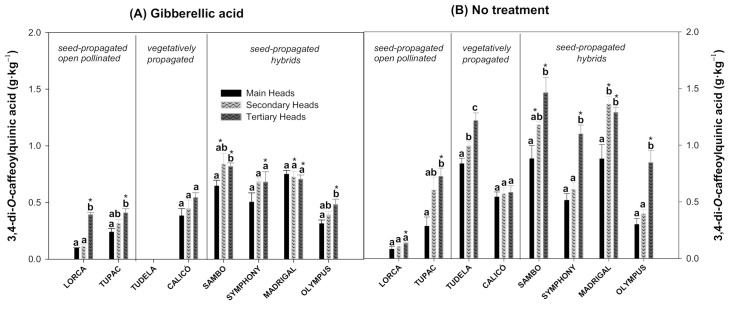
The 3,4-di-*O*-caffeoylquinnic acid content (g·kg^−1^) of globe artichoke cultivars influenced by flower head order (main, secondary, and tertiary heads) and treatment (gibberellic acid (**A**) or no treatment (**B**)). Data are the mean ± SE. Different letters show significant differences (*p* < 0.05 according to HSD Duncan’s test) among flower head orders for each artichoke cultivar and treatment. Significant differences between both treatments (*p* < 0.05 according to Student’s *t*-test) were expressed as * placed in each flower head order for each artichoke cultivar.

**Figure 5 foods-10-01813-f005:**
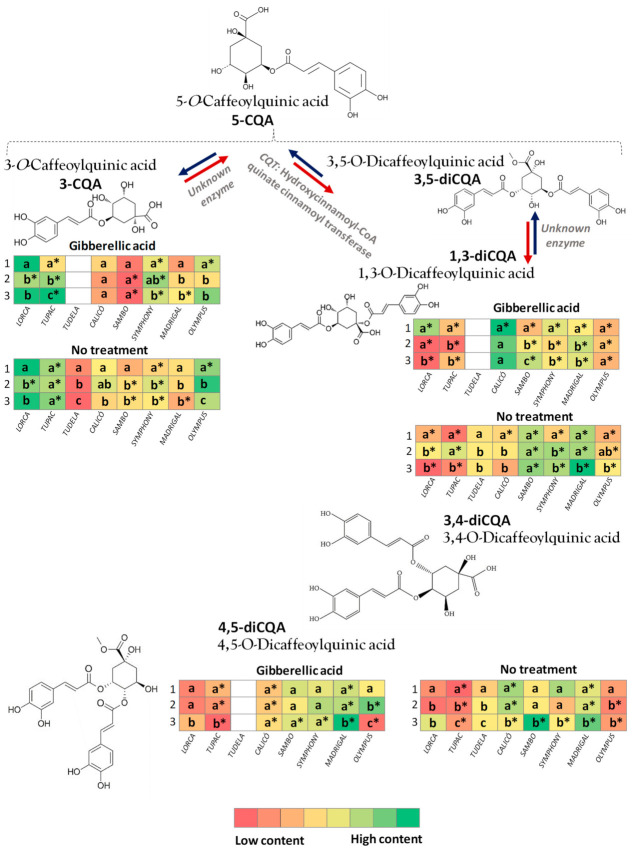
The 3-*O*-caffeoylquinnic acid (3-CQA), 1,3-di-*O*-caffeoylquinnic acid (1,3-diCQA), and 4,5-di-*O*-caffeoylquinnic acid (4,5-diCQA) content (mg·kg^−1^) of globe artichoke cultivars influenced by flower head order (main, secondary, and tertiary heads represented in the heatmap with numbers 1, 2, and 3) and treatment (gibberellic acid or no treatment). The heatmap shows the biosynthesis pathway of the three minor hydroxycinnamic acids from the major ones analysed. Data are the mean ± SE. Colours in the diagram represent the low or high content, ranging from red to green, respectively. Different letters within the same column show significant differences (*p* < 0.05 according to HSD Duncan’s test) among flower head orders for each artichoke cultivar and treatment. Significant differences between both treatments (*p* < 0.05 according to Student’s *t*-test) were expressed as * placed in each block for each artichoke cultivar and flower head order.

**Figure 6 foods-10-01813-f006:**
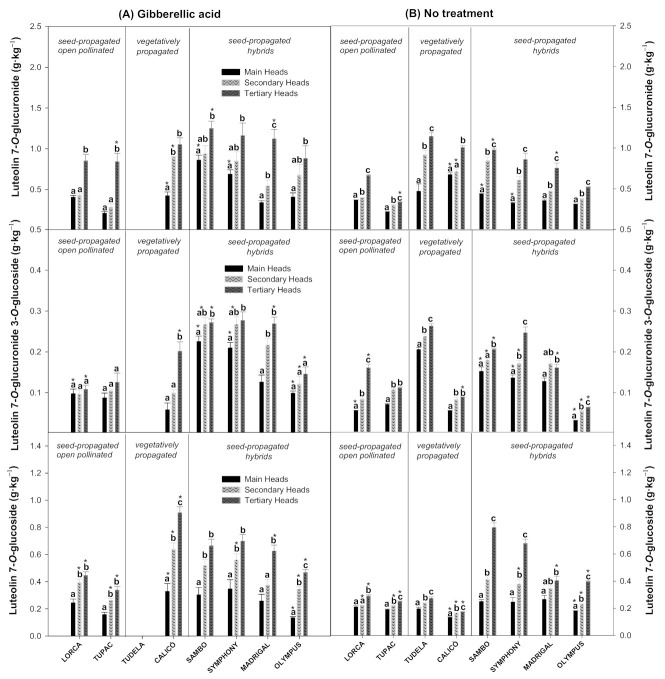
Luteolin 7-O-glucuronide, luteolin 7-O-glucuronide 3-O-glucoside, and luteolin 7-O-glucoside content (g·kg^−1^) of globe artichoke cultivars influenced by flower head order (main, secondary, and tertiary heads) and treatment (gibberellic acid (**A**) or no treatment (**B**)). Data are the mean ± SE. Different letters show significant differences (*p* < 0.05 according to HSD Duncan’s test) among flower head orders for each artichoke cultivar and treatment. Significant differences between both treatments (*p* < 0.05 according to Student’s *t*-test) were expressed as * placed in each flower head order for each artichoke cultivar.

**Table 1 foods-10-01813-t001:** Average flower head weight (g) of globe artichoke cultivars influenced by flower head order (main, secondary, and tertiary heads) and treatment (gibberellic acid or no treatment) ^1^.

	Gibberellic Acid			No Treatment		
	Main Heads	Secondary Heads	Tertiary Heads	Main Heads	Secondary Heads	Tertiary Heads
LORCA	240.37 ± 10.40c	175.76 ± 12.84b	122.57 ± 4.08a	273.82 ± 19.68c	160.48 ± 3.20b	123.17 ± 8.50a
TUPAC	280.99 ± 7.01b	178.65 ± 9.86a	167.15 ± 7.91a	306.38 ± 14.57c	179.52 ± 5.38b	148.61 ± 4.66a
TUDELA	-	-	-	209.39 ± 8.81c	180.34 ± 5.20b	156.33 ± 6.99a
CALICÓ	472.22 ± 32.75c	361.82 ± 23.25b	178.93 ± 13.25a *	496.62 ± 18.50c	333.26 ± 13.86b	261.69 ± 22.49a *
SAMBO	373.90 ± 20.96c *	300.25 ±18.86b *	189.75 ± 9.94a *	540.20 ± 13.58c *	393.21 ± 11.93b *	266.39 ± 18.76a *
SYMPHONY	243.13 ± 15.58c *	158.69 ± 8.24b	106.55 ± 6.83a	391.55 ± 21.64c *	196.34 ± 12.69b	121.33 ± 6.40a
MADRIGAL	297.83 ± 10.70c *	222.88 ± 10.17b *	155.40 ± 5.45a *	455.75 ± 32.62c *	279.19 ± 15.92b *	224.87 ± 7.44a *
OLYMPUS	254.83 ± 12.25c	206.98 ± 13.64b	138.59 ± 8.58a *	281.98 ± 11.76c	225.26 ± 7.04b	182.26 ± 12.33a *

^1^ Data are the mean ± SE. For each cultivar and treatment, different letters, within the same row, show significant differences at *p* < 0.05 among flower head orders, according to Tukey’s multiple range test. For each cultivar and flower head order, * shows significant differences at *p* < 0.05, according to Student’s *t*-test.

## Data Availability

Data have been addressed in the manuscript and in the Appendix A.

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
