# Peer review of "The Influence of Flower Head Order and Gibberellic Acid Treatment on the Hydroxycinnamic Acid and Luteolin Derivatives Content in Globe Artichoke Cultivars"

_foods, 2021, doi:10.3390/foods10081813_

Round 1
Reviewer 1 Report
Authors aimed to determine the phenolic composition in various flower heads of Cynara cardunculus plants. However, the identification and quantification of these metabolites can be considered insufficient based on the following facts (1-3):
1) Method of compound quantification is completely missing.
2) There are no evidences confirming the identity of the metabolites.
3) The sample chromatogram showing the separation of flavonoids (Supplementary Fig. S2B) should not be used for identification and quantification purposes: separation is insufficient, signal to noise ratio is too small, presence of unacceptable negative signals.
Author Response
Dear reviewer,
Thank you very much for your useful comments which have aid to improve our original manuscript. In the attached document you can find an itemed list of your comments and suggestions and the answer and modification performed in the revised manuscript according to your suggestions.

Reviewer 2 Report
The manuscript titled “The influence of flower head order and gibberellic acid treatment on the hydroxycinnamic acid and luteolin derivatives content in globe artichoke cultivars” aims to discuss the effect of treatments of different parts of artichoke with gibberellic acid to increase their phenolic content.
The paper is well presented/discussed, and relevant data is attached as SI. Abstract and Conclusions are supported by the presented work.
It was a pleasure to see scientific tools/solutions to leverage the dissemination of results aiming to optimise the plants quality (bioactive potential, through their improvement in phenolics).
In my opinion, the manuscript presents a high quality.
Author Response
Dear reviewer,
Thank you very much for your useful comments which could lead to publish our recently obtained results and thanks for your compliments.

Reviewer 3 Report
As far as I am concerned the manuscript is well written. By this I mean it is relevant and interesting, the text is readable and clear. The results are presented in a beaver and legible. Conclusions supported by evidence and refer to theses contained in the introduction. The subject area of research is significant in knowledge development. The introduction is interesting and correct.
Research well documented and described.
Conclusions result from the described research, well prepared. Rich supplementary materials.
Editing note:
In my opinion, the unit % should be written without spaces after the value - line 127, and next,
In g kg-1 units there should be a dot after g
Author Response
Dear reviewer,
Thank you very much for your useful comments which have aid to improve our original manuscript. In the attached document you can find the editing note with your comments and the answer and modification performed in the revised manuscript according to your suggestions

Round 2
Reviewer 1 Report
Taking into consideration the Reviewer’s opinion the authors have improved the manuscript where necessary.